# Spatio-temporal Representations of Uncertainty in Spiking Neural Networks

**Cristina Savin**
IST Austria
Klosterneuburg, A-3400, Austria
csavin@ist.ac.at

**Sophie Deneve**
Group for Neural Theory, ENS Paris
Rue d'Ulm, 29, Paris, France
sophie.deneve@ens.fr

## Abstract

It has been long argued that, because of inherent ambiguity and noise, the brain needs to represent uncertainty in the form of probability distributions. The neural encoding of such distributions remains however highly controversial. Here we present a novel circuit model for representing multidimensional real-valued distributions using a spike based spatio-temporal code. Our model combines the computational advantages of the currently competing models for probabilistic codes and exhibits realistic neural responses along a variety of classic measures. Furthermore, the model highlights the challenges associated with interpreting neural activity in relation to behavioral uncertainty and points to alternative population-level approaches for the experimental validation of distributed representations.

Core brain computations, such as sensory perception, have been successfully characterized as probabilistic inference, whereby sensory stimuli are interpreted in terms of the objects or features that gave rise to them [1, 2]. The tenet of this Bayesian framework is the idea that the brain represents uncertainty about the world in the form of probability distributions. While this notion seems supported by behavioural evidence, the neural underpinnings of probabilistic computation remain highly debated [1, 2]. Different proposals offer different trade-offs between *flexibility*, i.e. the class of distributions they can represent, and *speed*, i.e. how fast can the uncertainty be read out from the neural activity. Given these two dimensions, we can divide existing models in two main classes.

The first set, which we will refer to as *spatial* codes, distributes information about the distribution across neurons; the activity of different neurons reflects different values of an underlying random variable (alternatively, it can be viewed as encoding parameters of the underlying distribution [1, 2]). Linear probabilistic population codes (PPCs) are a popular instance of this class, whereby the log-probability of a random variable can be linearly decoded from the responses of neurons tuned to different values of that variable [3]. This encoding scheme has the advantage of speed, as uncertainty can be decoded in a neurally plausible way from the quasi-instantaneous neural activity, and reproduces aspects of the experimental data. However, these benefits come at the price of flexibility: the class of distributions that the network can represent needs to be highly restricted, otherwise the network size scales exponentially with the number of variables [1].

This limitation has lead to a second class of models, which we will refer to as *temporal* codes.These use stochastic network dynamics to sample from the target distribution [4, 1]. Existing models from this class assume that the activity of each neuron encodes a different random variable; the network explores the state space such that the time spent in any particular state is proportional to its probability under the distribution [4]. This representation is exact in the limit of infinite samples. It has several important computational advantages (e.g. easy marginalization, parameter learning, linear scaling of network size with the number of dimensions) and further accounts for trial-to-trial variability in neural responses [1]. These benefits come at the cost of sampling time: a fair representation of the underlying distribution requires pooling over several samples, i.e. integrating neural activity over time. Some have argued that this feature makes sampling unfeasibly slow [2].

Here we show that it is possible to construct *spatio-temporal* codes that combine the best of both worlds. The core idea is that the network activity evolves through recurrent dynamics such that samples from the posterior distribution can be linearly decoded from the (quasi-)instantaneous neural responses. This distributed representation allows several independent samples to be encoded simultaneously, thus enabling a fast representation of uncertainty that improves over time. Computationally, our model inherits all the benefits of a sampling-based representation, while overcoming potential shortcomings of classic temporal codes. We explored the general implications of the new coding scheme for a simple inference problem and found that the network reproduces many properties of biological neurons, such as tuning, variability, co-variability and their modulation by uncertainty. Nonetheless, these single or pairwise measures provided limited information about the underlying distribution represented by the circuit. In the context of our model, these results argue for using decoding as tool for validating distributed probabilistic codes, an approach which we illustrate with a simple example.

# 1    A distributed spatio-temporal representation of uncertainty

The main idea of the representation is simple: we want to approximate a real-valued $D$-dimensional distribution $P(\mathbf{x})$ by samples generated by $K$ independent chains implementing Markov Chain Monte Carlo (MCMC) sampling [5], $\mathbf{y}(t) = \{\mathbf{y}_k(t)\}_{k=1...K}$, with $\mathbf{y}_k \sim P(\mathbf{x})$ (Fig. 1). To this aim, we encode the stochastic trajectory of the chains in a population of $N$ spiking neurons ($N > KD$), such that $\mathbf{y}(t)$ is linearly decodable from the neural responses. In particular, we adapt a recently proposed coding scheme for representing time-varying signals [6] and construct stochastic neural dynamics such that samples from the target distribution can be obtained by a linear mapping of the spikes convolved with an epsp-like exponential kernel (Fig. 1a):

$$\hat{\mathbf{y}}(t) = \mathbf{\Gamma} \cdot \mathbf{r}(t) \tag{1}$$

where $\hat{\mathbf{y}}(t)$ denotes the decoded state of the $K$ MCMC chains at time $t$ (of size $D \times K$), $\mathbf{\Gamma}$ is the decoding matrix[1] and $\mathbf{r}$ is the low-pass version of the spikes $\mathbf{o}$, $\tau_V \dot{r}_i = -r_i + o_i$.

To facilitate the presentation of the model, we start by constructing recurrent dynamics for sampling a single MCMC chain, which we then generalise to the multi-chain scenario. Based on these network dynamics, we implement probabilistic inference in a linear Gaussian mixture, which we use in Section 2 to investigate the neural implications of the code.

**Distributed MCMC sampling**

As a starting point, consider the computational task of representing an arbitrary temporal trajectory (the gray line in Fig. 1b) as the linear combination of the responses of a set of neurons (one can think of this as an analog-to-digital conversion of sorts). If the decoding weights of each neuron points in a different direction (colour coded), then the trajectory could be efficiently reconstructed by adding the proper weight vectors (the local derivative of the trajectory) at just the right moment. Indeed, recent work has shown how to construct network dynamics enabling the network to track a trajectory as closely as possible [6]. To achieve this, neurons use a greedy strategy: each neuron monitors the current prediction error (the difference between the trajectory and its linear decoding from the spikes) and spikes only when its weight vector points in the right direction. When the decoding weights of several neurons point the same way (as in Fig. 1a), they compete to represent the signal via recurrent inhibition:[2] from the perspective of the decoder, it does not matter which of these neurons spikes next, so the actual population responses depend on the previous spike history, initial conditions and intrinsic neural noise.[3] As a result, spikes are highly irregular and look 'random' (with Poisson-like statistics), even when representing a constant signal. While competition is an important driving force for the network, neurons can also act cooperatively – when the change in the signal is larger than the contribution of a single decoding vector, then several neurons need to spike together to represent the signal (e.g. response to the step in Fig. 1a).

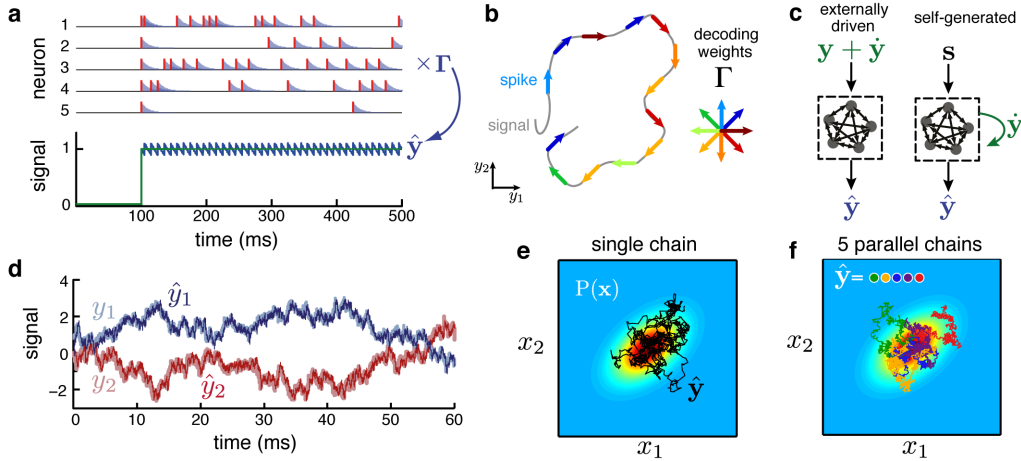

Figure 1: Overview of the model. **a.** We assume a linear decoder, where the estimated signal $\hat{y}$ is obtained as a weighted sum of neural responses (exponential kernel, blue). **b.** When the signal is multidimensional, different neurons are responsible for encoding different directions along the target trajectory (gray). **c.** Alternative network architectures: in the externally-driven version the target trajectory is given as an external input, whereas in the self-generated case it is computed via slow recurrent connections (green arrow); the input **s** is used during inference, when sampling from $P(\mathbf{x}|\mathbf{s})$. **d.** Encoding an example MCMC trajectory in the externally-driven mode. Light colours show ground truth; dark colours the decoded signal. **e.** Single-chain samples from a multivariate distribution (shown as colormap) decoded from a spiking network; trajectory subsampled by a factor of 10 for visibility. **e.** Decoded samples using 5 chains (colors) and a fifth of the time in **e**.

Formally, the network dynamics minimise the squared reconstruction error, $(\mathbf{y} - \hat{\mathbf{y}})^2$, under certain constraints on mean firing rate which ensure the representation is distributed (see Suppl. Info.). The resulting network consists of spiking neurons with simple leaky-integrate-and-fire dynamics, $\dot{\mathbf{V}} = -\frac{1}{\tau_v}\mathbf{V} - \mathbf{W}\mathbf{o} + \mathbf{I}$, where $\dot{\mathbf{V}}$ denotes the temporal derivative of $\mathbf{V}$, the binary vector $\mathbf{o}$ denotes the spikes, $o_i(t) = \delta$ iff $V_i(t) > \Theta_i$, $\tau_v$ is the membrane time constant (same as that of the decoder), the neural threshold is $\Theta_i = \sum_j \Gamma_{ij}^2 + \lambda$ and the recurrent connections, $\mathbf{W} = \mathbf{\Gamma}^T\mathbf{\Gamma} + \lambda \cdot \mathbb{I}$, can be learned by STDP [8], where $\lambda$ is a free parameter controlling neural sparseness. The membrane potential of each neuron tracks the component of the reconstruction error along the direction of its decoding weights. As a consequence, the network is balanced (because the dynamics aim to bring the reconstruction error to zero) and membrane potentials are correlated, particularly in pairs of neurons with similar decoding weights [7] (see Fig. 2c).

In the traditional form, which we refer to as the 'externally-driven' network (Fig. 1c), information about the target trajectory is provided as an external input to the neurons: $\mathbf{I} = \mathbf{\Gamma}^T \cdot (1/\tau_v \mathbf{y} + \dot{\mathbf{y}})$. In our particular case, this input implements a particular kind of MCMC sampling (Langevin). Briefly, the sampler involves stochastic dynamics driven by the gradient of $\log P(\mathbf{y})$, with additive Gaussian noise [5] (see Suppl.Info. for implementation details). Hence, the external input is stochastic $I = \mathbf{\Gamma}^T \cdot (1/\tau_v \mathbf{y} + F(\mathbf{y}) + \boldsymbol{\epsilon})$, where $F(\mathbf{y}) = \nabla \log P(\mathbf{y})$, and $\boldsymbol{\epsilon}$ is $D$-dimensional white independent Gaussian noise. Using our network dynamics, we can encode the MCMC trajectory with high precision (Fig. 1d). Importantly, because of the distributed representation, the integration window of the decoder does not restrict the frequency content of the signal. The network can represent signals that change faster than the membrane time constant (Fig. 1a, d).

To construct a viable biological implementation of this network, we need to embed the sampling dynamics within the circuit ('self-generated' architecture in Fig. 1c). We achieved this by approximating the current $\mathbf{I}$ using the decoded signal $\hat{\mathbf{y}}$ instead of $\mathbf{y}$. This results in a second recurrent input to the neurons, $\hat{\mathbf{I}} = \mathbf{\Gamma}^T \cdot (1/\tau_v \, \hat{\mathbf{y}} + F(\hat{\mathbf{y}}) + \boldsymbol{\epsilon})$. While this is an approximation, we found it does not affect sampling quality in the parameter regime when the encoding scheme itself works well (see example dynamics in Fig. 1e).

Such dynamics can be derived for any distribution from the broad class of product-of-(exponential-family) experts [9], with no restrictions on $D$; for simplicity and to ease visualisation, here we focus on the multivariate Gaussian case and restrict the simulations to bivariate distributions ($D = 2$). For a Gaussian distribution with mean $\boldsymbol{\mu}$ and covariance $\boldsymbol{\Sigma}$, the resulting membrane potential dynamics are linear:[4]

$$\frac{\partial \mathbf{V}}{\partial t} = -\frac{1}{\tau_{\mathrm{v}}} \mathbf{V} - \mathbf{W}^{\mathrm{fast}} \mathbf{o} + \mathbf{W}^{\mathrm{slow}} \mathbf{r} + \mathbf{D} + \boldsymbol{\Gamma}^{\mathrm{T}} \boldsymbol{\epsilon} \tag{2}$$

where $\mathbf{o}$ denotes the spikes, $\mathbf{r}$ is a low-passed version of the spikes. The connections $\mathbf{W}^{\mathrm{fast}}$ correspond to the recurrent dynamics derived above, while the slow[5] connections, $\mathbf{W}^{\mathrm{slow}} = \frac{1}{\tau_{\mathrm{slow}}} \cdot \boldsymbol{\Gamma}^{\mathrm{T}} \left( \mathbb{I} - \boldsymbol{\Sigma}^{-1} \right) \boldsymbol{\Gamma}$ (e.g. NMDA currents) and the drift term $\mathbf{D} = \frac{1}{\tau_{\mathrm{slow}}} \boldsymbol{\Gamma}^{\mathrm{T}} \boldsymbol{\Sigma}^{-1} \boldsymbol{\mu}$ correspond to the deterministic component of the MCMC dynamics[6] and $\boldsymbol{\epsilon}$ is white independent Gaussian noise (implemented for instance by a small chaotic subnetwork appropriately connected to the principal neurons). In summary, relatively simple leaky integrate-and-fire neurons with appropriate recurrent connectivity are sufficient for implementing Langevin sampling from a Gaussian distribution in a distributed code. More complex distributions will likely involve nonlinearities in the slow connections (possibly computed in the dendrites) [10].

**Multi-chain encoding: instantaneous representation of uncertainty**

The earliest proposal for sampling-based neural representations of uncertainty suggested distributing samples either across neurons or across time [4]. Nonetheless, all realisations of neural sampling use the second solution. The reason is simple: when equating the activity of individual neurons (either voltage or firing rate) to individual random variables, it is relatively straightforward to construct neural dynamics implementing MCMC sampling. It is less clear what kind of neural dynamics would generate samples in several neurons at a time. One naive solution would be to construct several networks that each sample from the same distribution in parallel. This however seems to unavoidably entail a 'copy-pasting' of all recurrent connections across different circuits, which is biologically unrealistic. Our distributed representation, in which neurons jointly encode the sampling trajectory, provides a potential solution to this problem. In particular, it allows several chains to be embedded in a single network.

To extend the dynamics to a multi-chain scenario, we imagine an auxiliary probability distribution over $K$ random variables. We want each to correspond to one chain, so we take them to be independent and identically distributed according to $\mathrm{P}(\mathbf{x})$. Since the sampling dynamics derived above do not restrict the dimensionality of the underlying distribution, we can use them to sample from this $D \times K$-dimensional distribution instead. For the example of a multivariate normal, for instance, we would now sample from another Gaussian, $\mathrm{P}\left(\mathbf{x}^{*K}\right)$, with mean $\boldsymbol{\mu}^{*K}$ ($K$ repetitions of $\boldsymbol{\mu}$) and co-variance $\boldsymbol{\Sigma}^{*K}$, a block-diagonal matrix, obtained by $K$ repetitions of $\boldsymbol{\Sigma}$. In general, the multi-chain trajectory can be viewed as just another instance of MCMC sampling, where the encoding scheme guarantees that the signals across different chains remain independent. What may change, however, is the interpretability of neural responses in relation to the underlying encoded variable. We show that under mild assumptions on the decoding matrix $\boldsymbol{\Gamma}$, the main features of single and pairwise responses are preserved (see below and Suppl.Info. Sec.4).

Fig. 1f shows an example run for multi-chain sampling from a bivariate Gaussian. In a fifth of the time used in the single-chain scenario (Fig. 1e), the network dynamics achieves a similar spread across the state space, allowing for a quick estimation of uncertainty (see also Suppl.Info. 2). For a certain precision of encoding (determined by the size of the decoding weights $\boldsymbol{\Gamma}$) and neural sparseness level, $N$ scales linearly with the dimensionality of the state space $D$ and the number of simultaneously encoded chains $K$. Thus, our representation provides a convenient trade-off between the network size and the speed of the underlying computation. When $N$ is fixed, faster sampling requires either a penalty on precision, or increased firing rates ($N \gg D$). Overall, the coding scheme allows for a linear trade-off between speed and resources (either neurons or spikes).

## 2 Neural implications

To investigate the experimental implications of our coding scheme, we assumed the posterior distribution is centred around a stimulus-specific mean (a set of $S = 12$ values, equidistantly distributed on a circle of radius 1 around the origin, see black dots in Fig. 3a), with a stimulus independent covariance parametrizing the uncertainty about $\mathbf{x}$. This kind of posterior arises e.g. as a result of inference in a linear Gaussian mixture (since the focus here is not on a specific probabilistic model of the circuit function, we keep the computation very basic, see Suppl. Info. for details). It allows us quantify the general properties of distributed sampling in terms of classic measures (tuning curves, Fano factors, FF, cross-correlogram, CCG, and spike count correlations, $r_{sc}$) and how these change with uncertainty.

Since we found that, under mild assumptions for the decoding matrix $\mathbf{\Gamma}$, the results are qualitatively similar in a single vs. a multi-chain scenario (see Suppl. Info.), and to facilitate the explanation, the results reported in the main text used $K = 1$.

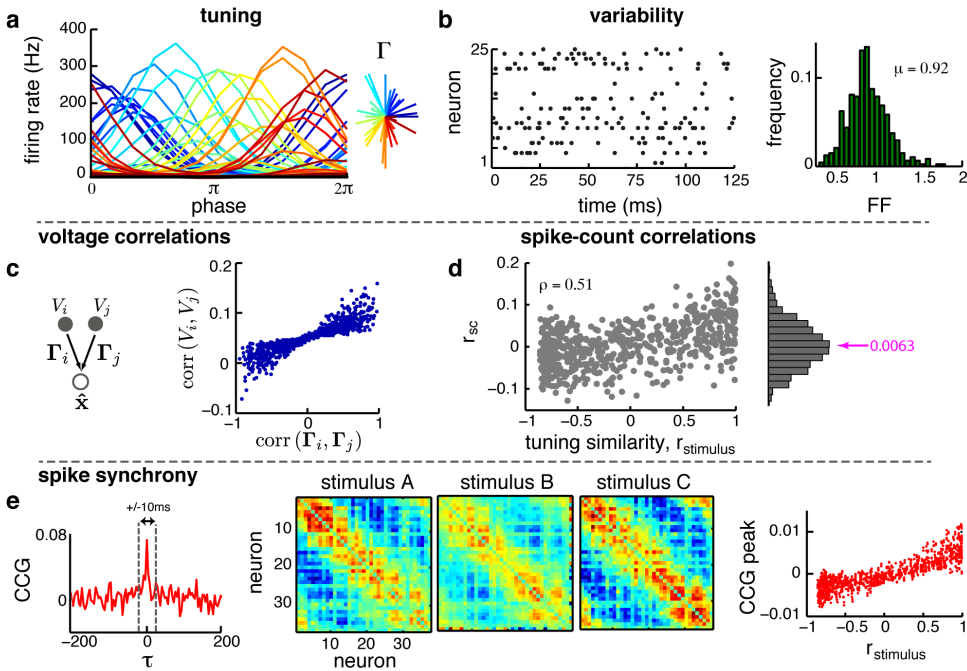

Figure 2: Our model recapitulates several known features of cortical responses. **a.** Mean firing rates as a function of stimulus, for all neurons ($N = 37$); color reflects the phase of $\mathbf{\Gamma}_i$ (right). **b.** The network is in an asynchronous state. Left: example spike raster. Right: Fano factor distribution. **c.** Within-trial correlations in membrane potential for pairs of neurons as a function of the similarity of their decoding weights. **d.** Spike count correlations (averaged across stimuli) as a function of the neurons' tuning similarity. Right: distribution of $r_{sc}$, with mean in magenta. **e** We use cross-correlograms (CCG) to asses spike synchrony. Left: CCG for an example neuron. Middle: Area under the peak $\pm10$ms (between the dashed vertical bars) for all neuron pairs for 3 example stimuli; neurons ordered by $\mathbf{\Gamma}_i$ phase. Right: the area under CCG peak as a function of tuning similarity.

### a. The neural dynamics are consistent with a wide range of experimental observations

First, we measured the mean firing rate of the neurons for each stimulus (averaged across 50 trials, each 1s long). We found that individual neurons show selectivity to stimulus orientations, with bell-shaped tuning curves, reminiscent of e.g. the orientation-tuning of V1 neurons (Fig. 2a). The inhomogeneity in the scale of the responses across the population is a reflection of the inhomogeneities in the decoding matrix $\mathbf{\Gamma}$.[7]

Neural responses were asynchronous, with irregular firing (Fig. 2b), consistent with experimental observations [11, 12]. To quantify neural variability, we estimated the Fano factors, measured as the ratio between the variance and the mean of the spike counts in different trials, $FF_i = \sigma_{f_i}^2 / \mu_{f_i}$. We found that the Fano factor distribution was centered around 1, a signature of Poisson variability. This observation suggests that the sampling dynamics preserve the main features of the distributed code described in Ref. [6]. Unlike the basic model, however, here neural variability arises both because of indeterminacies, due to distributed coding, and because of 'true' stochasticity, owed to sampling. The contribution of the latter, which is characteristic of our version, will depend on the underlying distribution represented: when the distribution is highly peaked, the deterministic component of the MCMC dynamics dominates, while the noise plays an increasingly important role the broader the distribution.

At the level of the membrane potential, both sources of variability introduce correlations between neurons with similar tuning (Fig. 2c), as seen experimentally [13]: the first because the reconstruction error acts as a shared latent cause, the second because the stochastic component –which was independent in the $y$ space– is mapped through $\mathbf{\Gamma}^{\mathrm{T}}$ in a distributed representation (see Eq. 2). While the membrane correlations introduced by the first disappear at the level of the spikes [7], the addition of the stochastic component turns out to have important consequences for the spike correlations both on the fast time scale, measured by CCG, and for the across-trial spike count covariability, measured by the noise correlations, $r_{\mathrm{sc}}$.

Fig. 2e shows the CCG of an example pair of neurons, with similar tuning; their activity synchronizes on the time scale of few milliseconds. In more detail, our CCG measure was normalised by first computing the raw cross-correlogram (averaged across trials) and then subtracting a baseline obtained as the CCG of shuffled data, where the responses of each neuron come from a different trial. The raw cross-correlogram for a time delay, $\tau$, $\mathrm{CCG}(\tau)$ was computed as the Pearsons correlation of the neural responses, shifted in time time by $\tau$.[8] At the level of the population, the amount of synchrony (measured as the area under the CCG peak $\pm 10$ms) was strongly modulated by the input (Fig. 2e, middle), with synchrony most prominent in pairs of neurons that aligned with the stimulus (not shown). This is consistent with the idea that synchrony is stimulus-specific [14, 15].

We also measured spike count correlation (the Pearsons correlation coefficient of spike counts recorded in different trials for the same stimulus) and found they depend on the selectivity of the neurons, with positive correlations for pairs of neurons with similar tuning (Fig. 2d), as seen in experiments [16]. The overall distribution was broad, with a small positive mean (Fig. 2d), as in recent reports [11, 12]. Taken together, these results suggest that our model qualitatively recapitulates the basic features of cortical neural responses.

**b. Uncertainty modulates neural variability and covariability**
We have seen that sampling introduces spike correlations, not seen when encoding a deterministic dynamical system [7]. Since stochasticity seems to be key for these effects, this suggests uncertainty should significantly modulate pairwise correlations. To confirm this prediction, we varied the covariance structure of the underlying distribution for the same circuit (Fig. 3a; the low variance condition corresponds to baseline measures reported above) and repeated all previous measurements. We found that changes in uncertainty leave neuronal tuning invariant (Fig. 3b, not surprisingly since the mean firing rates reflect the posterior mean). Nonetheless, increasing uncertainty had significant effects on neural variability and co-variability.

Fano factors increased for broader distributions (Fig. 3b), congruent with the common observation of the stimulus quenching response variability in experiments [17]. Second, we found a slower component in the CCG, which increased with uncertainty (Fig. 3e), as in the data [15]. Lastly, the dependence of different spike correlation measures on neural co-tuning increased with uncertainty (Fig. 3c, d). In particular, neurons with similar stimulus preferences increased their synchrony and spike-count correlations with increasing uncertainty, consistent with the stimulus quenching response co-variability in neural data and increases in correlations at low contrast [17, 16].

Although we see a significant modulation of (co-)variability with changes in uncertainty, these measures provide limited information about the underlying distribution represented in the network. They can be used to detect changes in the overall spread of the distribution, i.e. the high vs. low-variance

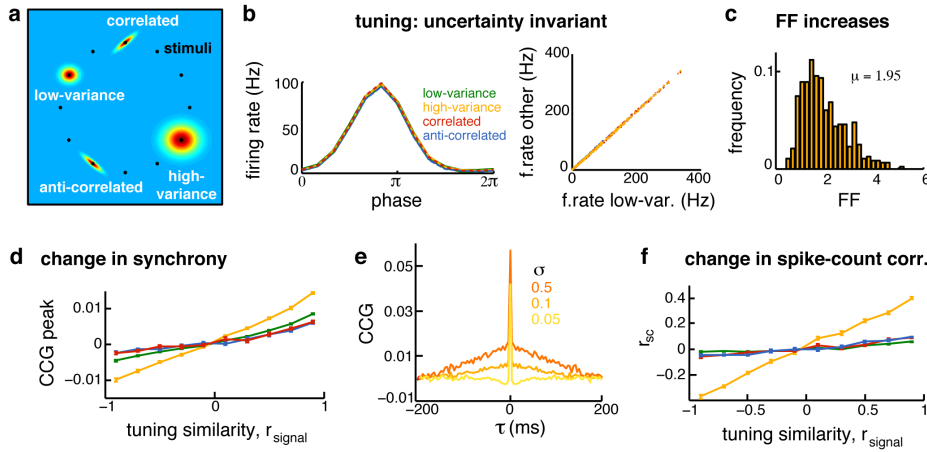

Figure 3: The effects of uncertainty on neural responses. **a.** Overview of different experimental conditions, posterior mean centred on different stimuli (black dots) with stimulus independent covariance shown for 4 conditions. **b.** Left: Tuning curves for an example neuron, for different conditions. Right: firing rate in the low variance vs. all other conditions, summary across all neurons; dots correspond to different neuron-stimulus pairs. **c.** Fano factor distribution for high-variance condition (compare Fig.2b). **d.** Area under CCG peak $\pm 10$ms as a function of the tuning similarity of the neurons, for different uncertainty conditions (colours as in **b**). **e.** Complete CCG, averaged across 10 neurons with similar tuning while sampling from independent bivariate Gaussians with different s.d. (0.1 for 'high variance'). **f.** Spike count correlations (averaged across stimuli) as a function of the tuning similarity of the neurons, for different uncertainty conditions.

condition look different at the level of pairwise neural responses. However, they cannot discriminate between distributions with similar spread, but very different dependency structure, e.g. between the correlated and anti-correlated condition (Fig. 3d, f; also true for FF and the slow component of the CCG, not shown). For this, we need to look at the population level.

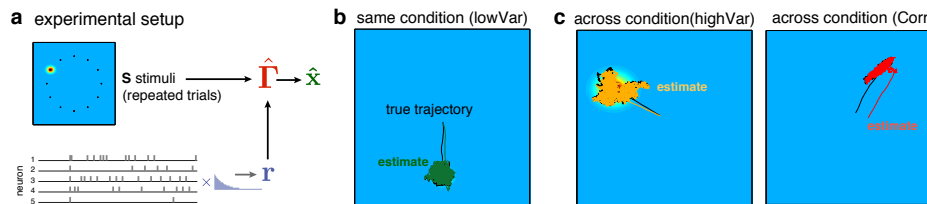

Figure 4: A decoding approach to study the encoding of uncertainty. **a.** In a low-variability condition we record neural responses for several repetitions of different stimuli (black dots); We estimated the decoding matrix by linear regression and used it to project the activity of the population in individual trials. **b.** The decoder captures well the underlying dynamics in a trial; ground-truth in black. **c.** The same decoder $\hat{\Gamma}$ can be used to visualise the structure of the underlying distribution in other conditions. Note the method is robust to a misalignment in initial conditions (red trace).

## c. Decoding can be used to assess neural representations of uncertainty

Since in a distributed representation single-neuron or pairwise measures tell us little about the dependency structure of the represented random variables, alternative methods need to be devised for investigating the underlying computation performed by the circuit. The representational framework proposed here suggests that linear decoding may be used for this purpose. In particular, we can record neural responses for a variety of stimuli and reverse-engineer the map between spikes and the relevant latent variables (or, if the assumed generative model is linear as here, the stimuli themselves). We can use the low-variance condition to get a reasonable estimate of the decoding matrix, $\hat{\Gamma}$ (since the underlying sampling dynamics are close to the posterior mean) and then use the decoder for visualising the trajectory of the network while varying uncertainty. As an illustration, we

use simple linear regression of the stimuli $s$ as a function of the neuron firing rates, scaled by $\tau_v$.[9] Although the recovered decoding weights are imperfect and the initial conditions unknown, the projections of the neural responses in single trials along $\hat{\Gamma}$ captures the main features of the underlying sampler, both in the low-variance and in other conditions (Fig. 4b, c).

## 3 Discussion

How populations of neurons encode probability distributions in a central question for Bayesian approaches to understanding neural computation. While previous work has shown that spiking neural networks could represent a probability over single real-valued variables [18], or the joint probability of many binary random variables [19], the representation of complex multi-dimensional real-valued distributions[10] remains less clear [1, 2]. Here we have proposed a new spatio-temporal code for representing such distributions quickly and flexibly. Our model relies on network dynamics which approximate the target distribution by several MCMC chains, encoded in the spiking neural activity such that the samples can be linearly decoded from the quasi-instantaneous neural responses. Unlike previous sampling-based codes [19], our model does not require a one-to-one correspondence between random variables and neurons. This separation between computation and representation is critical for the increased speed, as it allows multiple chains to be realistically embedded in the same circuit, while preserving all the computational benefits of sampling. Furthermore, it makes the encoding robust to neural damage, which seems important when representing behaviourally-relevant variables, e.g. in higher cortical areas. These benefits come at the cost of a linear increase in the number of neurons with $K$, providing a convenient trade-off between speed and neural resources. The speedup due to increases in network size is orthogonal to potential improvements in sampling efficiency achieved by more sophisticated MCMC dynamics, e.g. relying on oscillations [21] or non-normal stochastic dynamics [22], suggesting that distributed sampling could be made even faster by combining the two approaches.

The distributed coding scheme has important consequences for interpreting neural responses: since knowledge about the underlying distribution is spread across the population, the activity of single cells does not reflect the underlying computation in any obvious way. In particular, although the network did reproduce various properties of single neuron and pairs of neuron responses seen experimentally, we found that their modulation with uncertainty provides relatively limited information about the underlying probabilistic computation. Changes in the overall spread (entropy) of the posterior are reflected in changes in variability (Fano factors) and covariability (synchrony on the ms timescale and spike-count correlations across trials) of neural responses across the population, as seen in the data. Since these features arise due to the interaction between sampling and distributed coding, the model further predicts that the degree of correlations between a pair of neurons should depend on their functional similarity, and that the degree of this modulation should be affected by uncertainty. Nonetheless, the distributed representation occludes the structure of the underlying distribution (e.g. correlations between random variables), something which would have been immediately apparent in a one-to-one sampling code.

Our results reinforce the idea that population, rather than single-cell, responses are key to understanding cortical computation, and points to linear decoding as a potential analysis tool for investigating probabilistic computation in a distributed code. In particular, we have shown that we can train a linear decoder on spiking data and use it to reveal the underlying sampling dynamics in different conditions. While ours is a simple toy example, where we assume that we can record from all the neurons in the population, the fact that the signal is low-dimensional relative to the number of neurons gives hope that it should be possible to adapt more sophisticated machine learning techniques [23] for decoding the underlying trajectory traced by a neural circuit in realistic settings. If this could be done reliability on data, then the analysis of probabilistic neural computation would no longer be restricted to regions for which we have good ideas about the mathematical form of the underlying distribution, but could be applied to any cortical circuit of interest.[11] Thus, our coding scheme opens exciting avenues for multiunit data analysis.

## Footnotes

[1]The decoding matrix can be arbitrary.

[2]This competition makes spike correlations extremely weak in general [7].

[3]When $N \gg D$ there is a strong degeneracy in the map between neural responses and the signal, such that several different spike sequences yield the same decoded signal. In absence of internal noise, the encoding is nonetheless deterministic despite apparent variability.

[4]Since $F(\mathbf{x}) = \Sigma^{-1}\left(\mathbf{x} - \boldsymbol{\mu}\right)$, this results in a stochastic generalisation of the dynamics in [7].

[5]'Slow' marks the fact that the term depends on the low-passed neural output $\mathbf{r}$, rather than $\mathbf{o}$.

[6]Learning the connections goes beyond the scope of this paper; it seems parameter learning can be achieved using the plasticity rules derived for the temporal code, if these are local (not shown).

[7]The phase of the decoding weights was sampled uniformly around the circle, with an amplitude drawn uniformly from the interval $[0.005; 0.025]$.

[8]While this is not the most common expression for the CCG; we found it reliably detects synchronous firing across neurons; spikes discretised in 2ms bins.

[9] This requires knowledge of $\tau_v$ and, in a multi-chain scenario, a grouping of neural responses by chain preference. Proxies for which neurons should be decoded together are discussed in Suppl.Info. Sec.4.

[10] Such distribution arise in many models of probabilistic inference in the brain, e.g. [20].

[11] The critical requirement is to know (some of) the variables represented in the circuit, up to a linear map.

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
