[Supplementary Material]

# Spatio-temporal Representations of Uncertainty in Spiking Neural Networks
# – Supplementary Information –

**Cristina Savin**
IST Austria
Klosterneuburg, A-3400, Austria
csavin@ist.ac.at

**Sophie Deneve**
Group for Neural Theory, ENS Paris
Rue d'Ulm, 29, Paris, France
sophie.deneve@ens.fr

## 1   Distributed spike-based coding

The coding scheme has been derived elsewhere [1, 2] and is reproduced here for completeness. Unlike standard approaches, the model starts by specifying how information will be read out from the network. Given the specific decoder used for extracting information from the neural responses, we *derive* the network dynamics such as to yield the desired output. Specifically, we make the the Ansatz that the signal of interest $\mathbf{y}(t)$ can be decoded from the neuron responses linearly:

$$\hat{\mathbf{y}}(t) = \mathbf{\Gamma} \cdot \mathbf{r}(t) \tag{1}$$

where $\hat{\mathbf{y}}(t)$ is the network output, $\mathbf{\Gamma}$ is a fixed matrix corresponding to the decoding weights and $\mathbf{r}$ is the low-pass version of the spikes $\mathbf{o}$, $\dot{r}_i = -\frac{1}{\tau_V} r_i + o_i$.

Assuming that the signal to be encoded, $\mathbf{y}(t)$, is known[1], we want the neurons to spike as to minimize the $L_2$ error between the network output and this target, with a quadratic regularizer, parametrized by $\lambda$, which is meant to ensure sparseness and a more uniform utilisation of neural resources:

$$\mathcal{L} = (\mathbf{y} - \hat{\mathbf{y}})^{\mathrm{T}} (\mathbf{y} - \hat{\mathbf{y}}) + \lambda \cdot \mathbf{r}^{\mathrm{T}} \mathbf{r} \tag{2}$$

Note for notational convenience that we have dropped the explicit dependence of all vectors on time. To construct the network dynamics, we greedily minimize this loss function – a neuron $i$ emits an action potential whenever spiking would reduce the current error, i.e. if $\mathcal{L}|o_i = 1 < \mathcal{L}|o_i = 0$. Spelling out this constraint we obtain a condition for when the neuron should spike to reduce the loss $\mathcal{L}$:

$$(\mathbf{y} - \hat{\mathbf{y}})^{\mathrm{T}} (\mathbf{y} - \hat{\mathbf{y}}) + \lambda \sum_j r_j^2 \quad > \quad (\mathbf{y} - \hat{\mathbf{y}} - \mathbf{\Gamma}_i)^{\mathrm{T}} (\mathbf{y} - \hat{\mathbf{y}} - \mathbf{\Gamma}_i) + \lambda \left( 1 + \sum_j r_j^2 \right) \tag{3}$$

$$\mathbf{\Gamma}_i (\mathbf{y} - \hat{\mathbf{y}}) - \lambda r_i \quad > \quad 0.5 \cdot \left( \lambda + \mathbf{\Gamma}_i^{\mathrm{T}} \mathbf{\Gamma}_i \right) \tag{4}$$

To get this expression closer to the traditional form for leaky integrate-and-fire neurons, we can interpret the left side of this equation as the membrane potential of the neuron, $V_i = \mathbf{\Gamma}_i (\mathbf{y} - \hat{\mathbf{y}}) - \lambda r_i$, and the right side as the neural threshold, $\Theta_i = 0.5 \cdot \left( \lambda + \mathbf{\Gamma}_i^{\mathrm{T}} \mathbf{\Gamma}_i \right)$. To obtain the network dynamics for encoding the signal $\mathbf{y}$, we take the derivative of the membrane potential equation, and use the equation for $\dot{r}_i$ above. After some algebraic manipulation, this results in the final form of the network dynamics:

$$\dot{\mathbf{V}} = -\frac{1}{\tau_{\mathrm{v}}} \mathbf{V} - \mathbf{W} \mathbf{o} + \mathbf{\Gamma}^{\mathrm{T}} \left( \dot{\mathbf{y}} + \frac{1}{\tau_{\mathrm{v}}} \mathbf{y} \right) \tag{5}$$

where $\mathbf{W} = \mathbf{\Gamma}^{\mathrm{T}}\mathbf{\Gamma} + \lambda \cdot \mathbb{I}$ corresponds to the recurrent connections in the network.

When the signal to be represented is given, then the last term in this equation corresponds to an external input provided to the network, $I = \mathbf{\Gamma}^{\mathrm{T}}\left(\dot{\mathbf{y}} + \frac{1}{\tau_{\mathrm{v}}}\mathbf{y}\right)$. Alternatively, we can assume the evolution of y is described by a differential equation of the form $\dot{\mathbf{y}} = \frac{1}{\tau_{\mathrm{slow}}}F(\mathbf{y})$, where $F$ is an arbitrary function. In this case, rather than being provided as an external input the last term is computed in the circuit as $I = \mathbf{\Gamma}^{\mathrm{T}}\left(\frac{1}{\tau_{\mathrm{slow}}}F(\hat{\mathbf{y}}) + \frac{1}{\tau_{\mathrm{v}}}\hat{\mathbf{y}}\right)$. Note that this second scenario involves an approximation step, using the network output $\hat{\mathbf{y}}$ as a proxy for $\mathbf{y}$. While the success of this approximation depends on the specific function $F$, it seems to work well in practice provided that the parameters are such that the signal is well represented in the externally-driven version of the system. The linear version of the signal dynamics has been analyzed in detail previously [2]. The additional step required for distributed MCMC sampling to assume that the function $F$ is stochastic (Eq.2), and corresponds to the gradient of the Hamitonian as we explain in the following. For inference, the function $F$ is depends additionally on the sensory evidence.

## 2   Langevin sampling

Langevin sampling is a special case from the broad class of Hamiltonian Monte Carlo (HMC) methods [3], developed for efficient sampling in continuous spaces. The general approach can be summarised as following: to sample from a $D$-dimensional, real-valued distribution, $\mathrm{P}(\mathbf{x})$, we define a set of auxiliary variables,[2] $\mathbf{p}$, (of size $D$). We use these to construct an auxiliary distribution $\mathrm{P}(\mathbf{x}, \mathbf{p})$ from which we can sample from using standard Hamiltonian dynamics.

Formally, the random variables $\mathbf{x}$ and $\mathbf{p}$ are assumed to be independent, $\mathrm{P}(\mathbf{x}, \mathbf{p}) = \mathrm{P}(\mathbf{x}) \cdot \mathrm{P}(\mathbf{p})$; furthermore, the momentum variables are also independent and Gaussian, with zero mean, and unit variance, i.e. $\mathbf{p} \sim \mathcal{N}(\mathbf{p}; \mathbf{0}, \mathbb{I})$. To construct the dynamics, we need to compute the negative log-probability of the distribution, $H(\mathbf{x}, \mathbf{p}) = -\log \mathrm{P}(\mathbf{x}, \mathbf{p})$, which corresponds in physics terms to the total energy of the system; this can be decomposed into a potential energy term, $U(\mathbf{x}) = -\log \mathrm{P}(\mathbf{x})$, and a kinetic energy, $K(\mathbf{p}) = -\log \mathrm{P}(\mathbf{p}) = 0.5 \cdot \mathbf{p}^{\mathrm{T}} \cdot \mathbf{p}$ (as $\mathbf{x}$ and $\mathbf{p}$ are independent and $\mathbf{p}$ is Gaussian). These energies correspond to the following equations of motion:

$$\dot{x}_i \;=\; \frac{\partial H}{\partial p_i} = p_i \tag{6}$$

$$\dot{p}_i \;=\; \frac{\partial H}{\partial x_i} = \frac{\partial U}{\partial x_i} \tag{7}$$

Given these dynamics, the MCMC sampling proceeds by iteratively applying the following steps: we run the dynamics, starting from the current position $(\mathbf{x}, \mathbf{p})$, for a time $T$ and collect the final position $(\mathbf{x}^*, \mathbf{p}^*)$.[3] We use the final position as a proposal for the new state of the MCMC chain; this proposal is then accepted in the usual way, with a probability given as the ratio of the probabilities in the new vs. old state. The entire procedure is guaranteed to result in samples $\{\mathbf{x}\}$ distributed according to $\mathrm{P}(\mathbf{x})$. As a particularity of the Hamiltonian dynamics, the next state can be far from the original, yet has a very high probability of acceptance, ensuring efficient sampling. The Langevin dynamics are obtained in the limit of $T$ being very small (a single integration step).

One implementation-level complication of these methods is the fact that the method used for integrating the dynamics needs to be volume-preserving (see [3] for details). If using a simple leap-frogging algorithm to solve this problem, Langevin results in the following discrete update steps:

$$p_i \;\sim\; \mathcal{N}(p_i; 0, 1) \tag{8}$$

$$x_i^* \;=\; x_i - \frac{\eta^2}{2}\frac{\partial U}{\partial x_i}(\mathbf{x}) + \eta\, p_i \tag{9}$$

$$p_i^* \;=\; p_i - \frac{\eta}{2}\frac{\partial U}{\partial x_i}(\mathbf{x}) - \frac{\eta}{2}\frac{\partial U}{\partial x_i}(\mathbf{x}) \tag{10}$$

where $\eta$ is the integration step. The proposed state $x_i^*$, $p_i^*$ is accepted with probability:

$$\min\left\{1,\ \exp\left(-\left(U(\mathbf{x}^*)-U(\mathbf{x})-\frac{1}{2}\sum_i\left((p_i^*)^2-p_i^2\right)\right)\right)\right\}$$

If we ignore the acceptance step[4] (by making $\eta$ small enough such that the acceptance probability is very close to 1), and make the notational change $p_i \to \epsilon$ and $\nabla U = F$ then the dynamics reduce to the form used for the network:

$$x_i(t+1) = x_i(t) - \frac{\eta^2}{2}\,F_i(\mathbf{x}) + \eta\,\epsilon \tag{11}$$

with the noise term $\epsilon \sim \mathcal{N}(\epsilon; 0, 1)$.

## 3 Computational implication of a multi-chain encoding

Figure 1: We assume a bivariate normal posterior and compare the estimate of the mean and variance of $x_1$ (marginalising out $x_2$) as a function of the sampling time. We compare the estimates obtained by 1 (blue) or 25 (green) independent chains. The ground truth is shown in red; shaded regions show s.e.m. for 10 repetitions. The improvement in the multi-chain scenario is due partly to the trivial fact that there are more samples available at any specific time, but also because samples within a chain are correlated, whereas samples across chains are independent.

## 4 Network dynamics for probabilistic inference

We chose inference in a linear mixture with Gaussian noise as a simple example of probabilistic computation and used it to investigate the broad implications of distributed sampling on neural responses.[5] Briefly, the generative model assumes stimuli arise as a linear combination of $D$ underlying causes with multivariate Gaussian noise, $\mathbf{s} = \mathbf{A}\mathbf{x} + \boldsymbol{\eta}$, leading to the likelihood $P(\mathbf{s}|\mathbf{x}) = \mathcal{N}(\mathbf{s};\ \mathbf{A}\mathbf{x}, \boldsymbol{\Sigma}_\eta)$, where $\boldsymbol{\Sigma}_\eta$ is the noise covariance. We assume also a multivariate Gaussian prior, $P(\mathbf{x}) = \mathcal{N}(\mathbf{x}; \boldsymbol{\mu}_{\text{prior}}, \boldsymbol{\Sigma}_{\text{prior}})$, which results in a normally distributed posterior:

$$P(\mathbf{x}|\mathbf{s}) \propto P(\mathbf{s}|\mathbf{x}) \cdot P(\mathbf{x}) = \mathcal{N}\left(\mathbf{x}; \boldsymbol{\mu}_{\text{posterior}}, \boldsymbol{\Sigma}_{\text{posterior}}\right) \tag{12}$$

with parameters:

$$\boldsymbol{\mu}_{\text{posterior}} = \boldsymbol{\Sigma}_{\text{posterior}} \cdot \left(\boldsymbol{\Sigma}_{\text{prior}}^{-1} \cdot \boldsymbol{\mu}_{\text{prior}} + \mathbf{A}^{\text{T}} \cdot \boldsymbol{\Sigma}_{\text{noise}}^{-1} \cdot \mathbf{s}\right) \tag{13}$$

$$\boldsymbol{\Sigma}_{\text{posterior}}^{-1} = \boldsymbol{\Sigma}_{\text{prior}}^{-1} + \mathbf{A}^{\text{T}} \cdot \boldsymbol{\Sigma}_{\text{noise}}^{-1} \cdot \mathbf{A} \tag{14}$$

Replacing these posterior parameters in Eq. 3 from the main text, we obtain the network dynamics implementing inference in our generative model. The main difference is that the drift $\mathbf{D}$ is replaced

by two terms, a stimulus-dependent one, corresponding to a feedforward input to the circuit (main text, Fig.1c, 'self-generated'), $\mathbf{W}^{\mathrm{in}} = \frac{1}{\tau_{\mathrm{slow}}} \mathbf{\Gamma}^{\mathrm{T}} \mathbf{A}^{\mathrm{T}} \cdot \mathbf{\Sigma}_{\mathrm{noise}}^{-1} \cdot \mathbf{s}$, and another constant which reflects the prior, $\mathbf{W}^{\mathrm{prior}} = \frac{1}{\tau_{\mathrm{slow}}} \mathbf{\Gamma}^{\mathrm{T}} \mathbf{\Sigma}_{\mathrm{prior}}^{-1} \boldsymbol{\mu}_{\mathrm{prior}}$. The simple dependence on the stimulus is characteristic of a Gaussian likelihood, used in many classic models (e.g. changing the prior, we could get sparse coding [4]); other noise models may require a nonlinear influence of the stimulus $\mathbf{s}$ on the dynamics.

## 5 Neural tuning in a multi-chain scenario

Figure 2: **a.** Mean firing rates as a function of stimulus, for all neurons. **b.** Mechanics of stimulus encoding in the multi-chain scenario. Left: decoding weights corresponding to all chains for an example neuron (gray) and their vector sum (blue); Middle: tuning curves for 10 randomly selected example neurons, star on top marks the phase of $\sum_k \mathbf{\Gamma}_i^k$ (the neuron in the first panel is the same as left). Right: decoding weights for the tuning curves of the corresponding colour. **c.** Spike count correlations for an example stimulus. Neural index sorted by preferred chain (defined in text) and phase within that chain; black lines mark boundaries of the subpopulation preferring the 2nd chain. **d.** Spike count correlations (averaged across stimuli) as a function of the neurons' tuning similarity; mean of this distribution shown in magenta. **e.** Example CCG (averaged across trials for a single stimulus) for two example neuron pairs, the first pair with both neurons sharing the same chain preference, the second involving neurons with different chain preferences.

To quantify the properties of neural responses in the multi-chain we repeated the basic experiment in main text, Fig.2 for $K = 5$, using the same parameters. We found that neurons continue to exhibit stimulus tuning (Fig. 2a). Thus, the multi-chain network is similar to the single-chain case, but it appears more overcomplete. To better understand the reason for this tuning, we looked at individual

neurons and analysed their stimulus preference in relation to their decoding weights $\mathbf{\Gamma}_i$ (Fig. 2b). Since the decoding weights are initialised at random, each neuron represents different directions in $\mathbf{y}$ space for different chains; to quantify the relationship between these decoding weights and the neuron's stimulus preference we looked at the vector sum of the neuron's contributions in different chains (Fig. 2b). We found that this measure predicts reasonably well the tuning of the neuron as measured by average firing rates (this trend is true across the population; even when there are strong individual components in different directions, or when the sum vector is small compared to the individual components, see Fig. 2b, right). This finding suggests that realistic tuning curves can be obtained as long as the net contribution to the decoder $\mathbf{\Gamma}_i$ is not vanishing.[6]

The Fano factors did not differ significantly from the single-chain case (not shown). More interestingly, we found the spike-count correlogram exhibit a block diagonal structure (Fig. 2c). To reveal this structure, we had to reorder neuron indices by each neuron's preferred chain (the chain corresponding to the largest decoding vector, $k^* = \mathrm{argmax}_k |\mathbf{\Gamma}_i|$) and then by the neuron's phase preference within this chain. We found that the off-diagonal component is very close to zero all-round, suggesting that despite the significant across-chain overlap in $\mathbf{\Gamma}$, the activity of neurons preferring different chains is essentially decorrelated; in contrast, within-chain neural pairs recapitulate the correlation structure observed in the single chain scenario. Ultimately, these results mean that the spike-correlations are elevated in neurons with similar decoding prefereces in the $K \times D$ dimensional space in which the actual signal resides, i.e. the $K$ chains. Hence, these observations are consistent with our findings about the representation of a single chain.

When pooling correlations across all neuron pairs, we found they continue to express a dependence on tuning (inherited from the block-diagonal part), and a small positive mean (Fig. 2d). Fast time-scale correlations exhibited the same general patterns, with neurons within the same chain showing tuning-dependent synchrony and responses across chains being asynchronous (see Fig. 2e for a representative example; population averages not shown).

These findings are reminiscent of the experimental observation that pairs of neurons that are spatially near to each other show higher and tuning-dependent correlations, whereas at larger distances activities become uncorrelated [5, 6]. It is tempting to hypothesize that this locality in the correlation structure could reflect a topological organisation of neurons by preference to different chains. Such spatial clustering would be important for interpreting experimental data, especially in the context of decoding. In particular, one could use information about spatial distance and the presence (and tuning dependence) of correlations to isolate subsets of neurons likely to correspond to a single chain, which could then be used for the population analyses described in the main text.

## 6  Simulations details

The default parameter values used for all simulations were: $\tau_v = 20$ms for the membrane potential, $\lambda = 6.25/N^2$, $\eta = \frac{\tau_v}{\tau_{\text{slow}}} = 0.002$. We used a linear scaling of the network size with the dimensionality of the signal represented, $N = K \times 37$. For the single chain scenario, the decoding matrix was initialised at random, by sampling the orientation of the decoding vector uniformly on the unit sphere, then sampling its magnitude uniformly on the interval $[0.25, 1.25]$. For the multi chain results in Fig. 2, we generated the decoding matrix $\mathbf{\Gamma}$ as having a block diagonal structure, on the backdrop of weaker random across-chain weights (preserving the overall norm of the decoding vectors). This additional structure facilitates visualisation of the results but is not needed for the results. When assessing the neural implications of the scheme we used 50 trials, each lasting 1 second. To describe the effect the sensory input, we assumed a dependency such that the posterior mean tracks different $\mathbf{x}$ coordinates on a circle of radius 1[7] and manipulated the magnitude and structure of the correlations, as shown in Fig. 3a in the main text. When plotting the tuning curves, we used test stimuli distributed uniformly around the circle, identified by their phase. Lastly, for the results on decoding, we first simulated a sensory-like experiment in the low uncertainty condition (as before, 12 stimuli, 50 trials each, 1 second long), we convolved the observed spikes with an exponential kernel parametrized by the membrane potential $\tau_v$. We used simple linear regression

to infer a linear map $\hat{\Gamma}$ between these instantaneous firing rates and the known stimuli. This map was then used to visualise the responses of the same network on test data obtained under different uncertainty conditions.

## Footnotes

[1]This corresponds to the externally-driven network described in the main text; we relax this assumption later for the self-generated signal scenario.

[2]Due to a terminology borrowed from physics, these are called 'momentum' variables.

[3]In a physical analogy, this corresponds to kicking a ball on a surface shaped as $-\log \mathrm{P}$, starting from the current position, and tracking where it ends up after a fixed time $T$.

[4]Note that the acceptance step is a consequence of the discretisation used for the implementation of these dynamics on a computer, but does not concern a physical system evolving according to such dynamics.

[5]The focus here is the representation of the posterior rather than the underlying probabilistic computation.

[6]Hence, sampling uniformly the phase of $\mathbf{\Gamma}_i$ is not a good initialiser for the decoding weights when $K$ is large; here we added a modest bias term in the direction of one of the chains (different for different neurons ); on average the 'preferred' direction was 25% stronger than the rest.

[7]The spatial scale is arbitrary; it is determined by the overall magnitude of the decoding weights $\mathbf{\Gamma}$.