[Reviews · NeurIPS 2014]

Submitted by Assigned_Reviewer_19

This is a comprehensive review of neural sampling from probability distributions using spiking neurons, building on extensive work by Buesing, Maass, and others. Several illustrative examples are given.
Summary: This is a comprehensive review of neural sampling from probability distributions using spiking neurons, building on extensive work by Buesing, Maass, and others. Several illustrative examples are given.

Submitted by Assigned_Reviewer_29

The paper describes how a sampling approach for representing probability distributions with spiking neurons can be sped up using linear probabilistic population codes (PPCs). The paper first describes how a suitable choice of integrate and fire model can sample from a target distributions via a linear mapping of spikes. Then, it is argued that any product of exponentials distribution can fit within the dynamics of a PPC, and hence can be sampled akin to multi-chain MCMC. The main conclusion is that a single population of K neurons can thus sample K chains, in contrast to requiring K networks to sample K chains.
Most of the remainder of the paper is used to determine the neural implications of such scheme for encoding probability distributions, in terms of measures often used in neuroscience.

While the results are interesting, I find the paper unbalanced with respect to the NIPS audience: the PPC methods is described very briefly, and effectively only illustrated in figure 1f ever so briefly. The focus on neural implications (p5,p6 and p7) seems both too much and somewhat arbitrary: it demonstrates that the proposed encoding scheme with the particular chosen network topology is not incompatible with observations. In my opinion, the paper would have benefited from a more elaborate exposition of the results in section 1, including more elaborate demonstrations/graphs of the method.

I also have the concern that when reading correctly in the PPC paper by Ma et al (2006), there is no limitation in their PPC model that prevents multiple neurons from (similarly) independently computing posterior Gaussian distributions. I believe the distinction in the presented model comes from the particular use of spike-coding rather than rate-coding in the Ma et al paper. This is perhaps a point that should be elaborated more.

I will add that a number of papers on this topic (e.g. [6]) suffer from the same defect, where the technical details are very hard to extract from the paper (like the coding scheme in [6] as used in equation (1)); to my personal relief, other esteemed NIPS colleagues volunteered the same opinion unprompted.

Minor question: firing rates in figure 2a are very high for many biological neurons. To what degree does the scheme require such high firing rates?

Typo's:
p3 l142 each neurons -> each neuron
p3 l153 we can encoding -> we can encode
p3 l160 Q: shouldnt "x" be "y" in the equation (from line 159)?
p4 l188 the s sampling
p8 l386 in central question -> is a central question
Summary: Interesting approach that has a clear place at NIPS. However, the presentation is unbalanced and the computational contribution (sec 1) deserves more prominence and elaboration.

Submitted by Assigned_Reviewer_34

The authors present a theoretical model of neural population dynamics. Under their model, a noise-driven linear dynamical system is designed to sample a given distribution via hamiltonian monte carlo (following the work of (Boerlin et al, 2011)). At each timestep, the state of this system dictates the membrane potential of a population of neurons, by linear projection. The authors show in simulation that their model reproduces summary statistics characteristic of neural populations (in toy examples), and consider the model's implications upon neural variability and decoding.

The paper is generally well-written and has a high standard of presentation. The discussion is a bit dense at times, and I believe it could benefit from clearer statements of methods at times (in particular: how exactly was the method of (Boerlin et al, 2011) modified? Also: Was the decoding matrix learned in the simulations? What were the time constants?).

The form of the model (low dimensional dynamics projected linearly to modulate population response) is related to recent work oriented more toward statistical modeling and data analysis (for instance (Pfau et al, 2013), (Turaga et al, 2013), (Yu et al, 2009), (Ecker et al, 2014)). Some of the points of the paper (for instance, the statistical gains of redundancy implicit in projecting low-dimensional dynamics into a high-dimensional population, the importance of population-level modeling) appear as motivation in some of this literature.The success of such statistical models renders this more theoretical & "biophysically plausible" approach more timely, but also makes me wish that real data were shown in the paper.

Overall, I think the approach is interesting and, and the simulated experiments do show compelling/interesting features. However, I find it difficult to extrapolate from the toy examples shown to larger-scale models of population activity.
Summary: The authors present a sampling-based model of latent, low-dimensional population dynamics, and show that it broadly matches the characteristics of a neural population in a simplified example. Though polished and well-written, the paper suffers somewhat from an absence of real neural data and the "toy" nature of simulations.

Submitted by Assigned_Reviewer_42

The authors present a recurrent spiking neural network model for sampling from multivariate probability distributions through MCMC. A key advantage of their approach is to separate computation and representation: Samples of the target distribution are obtained from the network response via a linear decoder, i.e. neurons do not need to directly correspond to random variables. This gives rise to a rich distributed spatio-temporal code to represent the underlying distribution. Furthermore, the resulting flexibility in the network architecture allows to run multiple MCMC chains in parallel and, thus, to approximate the target distribution near-instantaneously.
The theory is complemented by computer simulations that (a) demonstrate the general feasibility of the approach, (b) illustrate how the network performs inference (in a simple generative model) and (c) compare response characteristics with findings from experimental neuroscience. Finally, the authors explore how the approach can be used to identify (properties of) the underlying distribution when the linear decoder is not known; an important step for an application to experimental data.

Overall, this is a good manuscript. It contributes several new ideas which are of interest to a broader research community. The manuscript is mostly very well written (particularly the introduction and discussion), the computer simulations and figures illustrate the network properties well. Finally, the work perfectly fits the scope of the conference.

Nonetheless, I have some critical remarks which reduced the otherwise positive impression.

(A) The manuscript suffers from missing or extra words which, in some cases, even affect the content. E.g. in lines 46, 47, 142, 153, 188, 223 and 386.

(B) The spike trains o_i are not defined in line 80. The reset mechanism (described in [6]) is not mentioned around line 140.

(C) The derivation is highly compressed and definitely demands reading of refs [6-8]. Also the SI contains relevant information (e.g. the generative model definition). Furthermore, I didn't find important parameters of the simulations (e.g. number of neurons and decoding matrix in Fig 1, lambda, tau_slow,...). This renders a verification of the results almost impossible. Some details to the simulations should be added to the SI.

(D) While the network supports a high degree of freedom in the recurrent weight matrices, connections are still symmetric (at least for the Gaussian case). Such limitations should be mentioned at some point.

(E) I was wondering if all equations are fully correct. I didn't delve into the details, so the following are just some points the authors might want the check. In line 141, shouldn't the summation run over the first index of the decoder? The sparseness parameter lambda changes the spike response but does not enter any other equation to account for this effect. Is this correct? Similarly, the time constant tau_slow that scales the recurrent weights and drift is not further specified. Can it be chosen arbitrarily, or is this tau_v?

In summary, I had the impression that a bit too many different research questions were put into this nine pages manuscript. This comes at the cost of reduced clarity of the individual parts. For instance, since the derivation rests upon some approximations, it would be helpful to explore the range of validity. Also the intriguing multi-chain sampler would have deserved a more comprehensive presentation. Still, I consider this manuscript an important and valuable contribution to the conference if the authors address some of the above issues, and I hope that (one or more) follow-up papers can investigate the presented ideas in greater detail.
Summary: This is a good manuscript that introduces some highly interesting ideas. It suffers from some deficits in the presentation; but I am optimistic that they can be fixed in a final version.
Author Feedback
Author rebuttal: We thank the reviewers for their feedback.
Detailed comments follow below.

R1:
Although in the introduction we briefly review previous codes used for representing probability distributions in neural networks, the main contribution of this paper is a *new coding scheme* that relies on a distributed representation of MCMC sampling; we further show that the proposed code is both computationally useful (can be used to represent arbitrary multidimensional distributions while making the decoding of uncertainty faster than previous approaches) and consistent with a range of experimental observations.

R2:
There seem to be a misunderstanding/terminological confusion when linking our work to (linear) PPCs: the two are similar in that both assume linear decodability, nonetheless the quantities being encoded are very different — in PPCS: log probability, here: samples from the underlying distribution.

As a result, the two coding schemes differ significantly in terms of flexibility:
we used a multivariate gaussian as an example, but there is nothing about the coding scheme forcing the distribution to be normal (the exact expression for the slow currents depends on the distribution, but otherwise things remain unchanged); **this is not true for PPCs** (either everything is gaussian or the variables are independent).

Beyond the computational advantages provided by a new coding scheme (which we quantified in the Suppl. Info.) its plausibility as a model of neural computation can only be ascertained by comparing it to data. If in a temporal code the link between sampling and neural responses (variability/covariability) is relatively straightforward, this is no longer the case in a distributed representation such as ours. This is why we chose to dedicate a substantial part of the text to the neural consequences of our coding scheme. The specific measures analysed were selected to cover the spectrum of experimental observations that competing models aim to explain.

Our findings emphasise the need for new analysis tools in investigating neural coding at the population level, that would rely on state-of-the-art machine learning. Because of this, we feel the paper should be of interest to the greater NIPS community.

We edited the text to clarify these issues.

We agree that the presentation of the methods is somewhat dense. This was due to length restrictions but also because the basic encoding framework had been described in great detail previously (refs.6-8). For completeness, we now include the full derivation of the dynamics in the Suppl. Info.

The firing rates are high because we wanted a high-precision representation, but used relatively small networks. The mean firing rates can be easily reduced by increasing N, with no qualitative differences.

Thank you for pointing out the typos, which we have now corrected.

R3
The focus of the current paper was the new coding scheme and its biological implementation at the level of a spiking neural network. We used the simplest possible model for inference because we wanted to illustrate the broad consequences of the code, irrespective of the computational details of a specific neural circuit. Nonetheless, it would be interesting to extend these results to more complex generative models, for instance describing computation in V1 (inference in a gaussian scale mixture a la Schwartz & Simoncelli/ Orban et al etc).

We do not have access to the type of experimental data required for directly testing the model; nonetheless we do show that the newly proposed coding scheme is qualitatively consistent with published data, at least at the level of single neuron and pairwise responses. We hope to focus on population-level analyses in future work.

A clarification on the general approach: the decoding matrix is fixed and the optimal network dynamics are derived as a function of $\Gamma$.
Recent extensions of the scheme do allow the decoding matrix to be learned as well, on a different time scale, but we do not cover this scenario here.

We edited the text to clarify these issues.

R4:
We agree that the presentation is somewhat dense and incomplete in places. To improve it, we have edited the main text, corrected the typos, and extended the Suppl. Info. to include a full derivation of the dynamics and a section listing the exact setting of all parameters used in the simulations.

R4 is correct in that W needs to be symmetric. Some recent developments from the Deneve lab have explored ways to relax this assumption in the context of the broad encoding scheme, nonetheless it is not clear to which extent these could be applicable in our case. We edited the text to clarify this point.

tau_slow is a separate parameter, describing the speed of the MCMC dynamics.